# Efficiency of an Alternative Physical Education Program for the Lower Grades of Elementary School Children

**DOI:** 10.3390/children10101657

**Published:** 2023-10-06

**Authors:** Jovan Marković, Saša Bubanj, Goran Šekeljić, Slobodan Pavlović, Marko Radenković, Dušan Stanković, Emilija Petković, Nikola Aksović, Oliver Radenković, Adem Preljević, Bojan Bjelica, Vladan Petrović, Šćepan Sinanović, Milena Tomović

**Affiliations:** 1Faculty of Pedagogy, University of Kragujevac, 31000 Užice, Serbia; sekeljic@pfu.kg.ac.rs (G.Š.); pavlovic@pfu.kg.ac.rs (S.P.); 2Faculty of Sport and Physical Education, University of Niš, 18000 Niš, Serbia; sasabubanj@fsfv.ni.ac.rs (S.B.); radenkom9@gmail.com (M.R.); dukislavujac@gmail.com (D.S.); emilija@fsfv.ni.ac.rs (E.P.); 3Faculty of Medicine, University of Belgrade, 11000 Belgrade, Serbia; 4Faculty of Sport and Physical Education, University of Priština, 38218 Leposavić, Serbia; nikola.aksovic@pr.ac.rs; 5Department of Biochemical Science and Sport, State University of Novi Pazar, 36300 Novi Pazar, Serbia; oradenkovic@np.ac.rs (O.R.); apreljevic@np.ac.rs (A.P.); 6Faculty of Physical Education and Sport, University of East Sarajevo, 71420 Pale, Bosnia and Herzegovina; bojan.bjelica@ffvis.ues.rs.ba; 7Faculty of Mechanical Engineering, University of Niš, 18000 Niš, Serbia; vladan.petrovic@masfak.ni.ac.rs; 8Milutin Milanković High Medical College of Professional Studies, 11000 Belgrade, Serbia; pomocnik@vmsmmilankovic.edu.rs; 9Sports Medicine Laboratory, Department of Physical Education and Sports Science, Aristotle University of Thessaloniki, 54124 Thessaloniki, Greece; milenatomovic83@gmail.com

**Keywords:** young school children, alternative physical education curriculum, motor abilities, effects, differences

## Abstract

(1) Introduction: This research was conducted with the aim of assessing whether an alternative physical education (PE) program could effectively substitute for the traditional PE curriculum and aid in accomplishing the essential goals and objectives of PE among younger elementary school children. (2) Materials and Methods: This longitudinal 6-month study included third-grade elementary school children of both genders, who were healthy, lived in urban areas, and were involved in an alternative PE program, as well as the regular PE classes. The sample of participants comprised 214 students, with 105 participants in the experimental group and 109 participants in the control group, who underwent pre- and post-intervention measurements. For the purposes of this study, 11 variables were applied to assess the parameters of growth and development, motor abilities, and physical and health education. MANCOVA and ANCOVA methods were used to determine the effects resulting from the alternative and regular PE programs, and differences between the groups, respectively. The data are reported as the mean and standard deviations, and were analyzed using the statistical package SPSS version 20.0 (SPSS Inc., Armonk, NY, USA). (3) Results: Based on the research results obtained for motor abilities and physical and health education, it was concluded that both the alternative and regular PE programs had positive effects in achieving the goals and objectives of PE, but without statistical significance at the multivariate and univariate levels. The differences between the two groups were found to be negligible (effect size, ES < 0.2). (4) Discussion: Sports and PE have distinct objectives and approaches. While sports emphasize competition and winning, PE aims to impart fundamental skills and knowledge, prioritizing inclusivity among younger students. Success in PE is measured by the number of students meeting these goals, potentially affecting both talented and struggling learners. (5) Conclusions: The results obtained from the conducted research indicate that both the alternative PE program and the regular PE program influence changes in motor abilities and physical and health education to a limited extent. The alternative PE program proposed in this study, with its modifications to the structure of existing, regular PE program, can fully replace the latter in schools that do not meet the required spatial and material standards.

## 1. Introduction

Schools and society are obligated to provide the necessary conditions for the implementation of educational plans and programs, including those related to physical education (PE) [1]. This entails fulfilling all the prerequisites and standards essential for achieving the objectives and tasks of the educational process, encompassing human resources, spatial arrangements, material provisions, and technical requirements [2]. To exert an optimal influence on students, it is imperative that PE instruction be tailored to the age and developmental capacities of the children being instructed [3]. The cornerstone of successful educational endeavors is, among other factors, the formulation and execution of a robust, lucid, concrete, and realistic teaching plan. The framework of the specific objectives within the curriculum for PE in the educational system of the Republic of Serbia has been classified into three distinct categories: biological, educational, and developmental [4]. They are interlinked, interdependent, and collectively realized.

The aim of the aforementioned PE instruction is “to contribute, through diverse and systematic motor activities in conjunction with other educational domains, to the holistic development of the student’s personality (cognitive, affective, motor), the enhancement of motor abilities, the acquisition, refinement, and application of motor skills, habits, and indispensable theoretical knowledge within the contexts of daily life and specific work conditions” [4] (p. 63).

Quality physical activity in elementary school has been shown to be important to various aspects of children’s biological growth, development, and health [5]. Furthermore, cognitive functions like memory and attention have demonstrated a strong correlation with higher levels of physical activity and overall physical fitness among children [6]. In addition to cognitive abilities, the presence of overweight in children is even more alarming. Additionally, the prevalence of childhood overweight is of even greater concern. Several studies have examined the impacts of physical activity during the elementary school years on children’s well-being [6,7,8,9,10], highlighting the health benefits of physical activity.

Owing to variations in the development of motor abilities among students of differing ages, divergences in the operational objectives for PE instruction emerge across the different grades [11]. The motor domain undergoes transformations over an extended temporal span, and is characterized by opportune intervals during which motor structures can be effectively altered. Notably marked increments in the development of coordination, speed, flexibility, balance, and explosive strength are distinctive characteristics of third-grade elementary school students. This factor substantiates the inclusion of teaching contents in curricula and operational tasks that support the cultivation of these motor skills [12].

Unfortunately, many schools in Serbia lack adequate conditions for conducting PE classes [13], in terms of infrastructure and the availability and quality of equipment and apparatus [14]. Studies [14,15,16,17,18] have pointed to the fact that this problem has persisted for many years, even decades, and PE classes are conducted under challenging conditions that fall significantly below health–hygiene and technical standards. Furthermore, there is still a lack of societal awareness and responsibility regarding the significance of addressing this problem.

Particularly alarming is the realization that the majority, and often all the available capacity of school sports facilities, is utilized by students from the fifth to eighth grades. This highlights the lack of resources schools have for conducting PE classes for younger students, and emphasizes the fact that tens of thousands of children in this age group either lack adequate or, often, any PE instruction. This issue is particularly pronounced from November to May, when adverse weather conditions hinder outdoor PE classes. Practice has confirmed that teachers working in such inadequate conditions lack clear guidance and instructions on how to adapt their lesson plans to the technical and material conditions at hand, while still fulfilling the prescribed objectives and tasks of PE classes.

The pertinent ministry should address this matter with greater seriousness, rather than consistently focusing on reforms that have either not been fully implemented or have proven ineffective in practice. Previous efforts have shown an absence of significant results and have avoided a larger investment in PE [19].

This alarming state of affairs prompted us to create and implement an alternative PE program for schools that lack adequate conditions for regular PE classes. This initiative aimed to address a significant issue within our educational system and propose an alternative PE program as a solution.

This study sought to investigate the impact of a 6-month alternative PE program, incorporating dance and gymnastics, on younger school children. Its primary objectives were to assess whether this program could effectively replace the current curriculum and contribute to fulfilling the essential PE goals and objectives for this age group. This study’s specific focus was on evaluating the effects of the alternative PE program on the motor abilities and physical and health education of third-grade elementary school children.

## 2. Materials and Methods

### 2.1. Subjects

Based on a pilot study with a smaller sample, we determined that the preliminary effect size values (Cohen’s d) ranged between 0.28 and 0.45. Using the pwr.*t*.test function from the R programming package (library pwr), we concluded that the minimum sample size was N = 102. This study encompassed 214 nine-year-old third-grade students from five elementary schools in Užice, all belonging to the same generation. The methodology employed parallel groups. The experimental group consisted of 105 students who followed the alternative PE program for the lower-elementary-school-grades children, while the control group, comprising 109 students, adhered to the regular PE curriculum for third-grade students.

Given that the environment, urban or rural, in which children reside can influence their physical development and abilities [20], this study exclusively selected participants from urban environments to ensure a more homogeneous assessment of the effects of the experimental factor, namely the alternative program.

No selection or restrictions were imposed based on nationality, socioeconomic status, education, cognitive and conative dimensions, or gender.

The core criteria for selecting the sample rested on the participants having complete results for both the pre-intervention and post-intervention measurements, regular attendance in PE classes (with absences not exceeding 10% during the school year), and a chronological age of 9 years (with a deviation of ±6 months).

Written permission from the parents/guardians of all the participants was obtained for their children’s inclusion in this study. The Teaching-Scientific Council of the Faculty of Pedagogy, University of Kragujevac, granted approval for this study (Reference No. 7-10/19), which was conducted in accordance with the principles of the Declaration of Helsinki.

### 2.2. Experimental Design

Prior to and following the 6-month intervention period (from 1st November to 1st May, when weather conditions prevent outdoor PE classes in the yard or playground), assessments were conducted on all the participants.

The experimental group took part in the alternative PE program, which replaced the standard school PE curriculum. Meanwhile, the control group adhered to the regular school PE curriculum and did not participate in any supplementary organized physical activities. Both PE programs were conducted twice a week, with each session lasting 45 min. The pre-intervention and post-intervention evaluations for both groups encompassed the measurement of anthropometric characteristics and motor skills.

Both the students and teachers willingly accepted all the obligations associated with each phase of the program’s implementation. The teachers voluntarily undertook all the responsibilities related to measurements (pre-intervention and experimental), tracking students’ work and attendance, documenting any changes in the curriculum, and fulfilling any other obligations to ensure the highest-quality execution of the alternative PE program in the lower grades of elementary school.

Between September 1st and 15th, the pre-intervention measurements were conducted for all the children participating in this study. The post-intervention measurement of students was carried out between 1st May and 1st June of the following year.

Additionally, a project coordinator was assigned to each school to assist during testing, including with the pre-intervention and post-intervention measurements, as well as on other occasions as recommended by the teachers.

### 2.3. Anthropometric Characteristics

The anthropometric attributes were evaluated in accordance with the guidelines set forth by the International Biological Program, encompassing measurements of height and mass. Height measurements were conducted with a Martin metal anthropometer (GPM, Bachenbülach, Switzerland), accurate to the nearest 0.5 cm. Body mass was ascertained with a digital weight scale, providing precision up to the nearest 0.1 kg.

### 2.4. Academic Achievement

Information regarding the students’ final grade point average in the subject of physical and health education at the conclusion of the academic year, ranging from 1 to 5, was extracted from school records after obtaining authorization from the principals of the elementary schools.

Throughout one semester, students were required to receive four grades to determine their final evaluation. The final assessment was categorized as follows: “Insufficient” (1) if the average was below 1.50, “Acceptable” (2) if the average was between 1.50 and 2.49, “Good” (3) if the average ranged from 2.50 to 3.49, “Very Good” (4) if the average fell between 3.50 and 4.49, and “Outstanding” (5) if the average of all the individual grades was 4.50 or higher.

### 2.5. Motor Skills Assessment

Motor skills were evaluated using the Eurofit test battery that encompasses eight motor tests. This standardized test battery was devised for school-age children.

The battery of motor tests is structured to be completed within 35 to 40 min, utilizing relatively simple equipment. The tests are conducted sequentially (the Flamingo balance test always initiates the process, while the 10 × 5 m shuttle run is invariably the final test) and are described below in their order of appearance. The Flamingo balance test (FBT) in s: The test commences after a practice attempt. Three trials are executed, and the longest balancing time on the beam is recorded. Hand tapping (HT) in s: The task entails completing twenty-five cycles. The recorded time is the shortest completion time measured in tens of seconds. Attempts where both plates are not touched are not counted. Seated forward bend (SFB) in cm: The result, in centimeters, signifies the grade or reach on the scale. If the distances reached by both hands differ, the average is computed. Standing broad jump (SBJ) in cm: The distance covered from the starting line to the nearest point of contact on landing is measured. The subject performs two consecutive jumps, and the better jump is taken. Incorrectly performed jumps are repeated. Handgrip strength (HS) in kg: The subject’s grip strength is assessed, with the result expressed in kilograms, rounded to 1 kg for accuracy. The stronger grip is noted. Sit-and-reach (S & R) in repetitions: the number of correctly executed sit-ups, with the elbows touching the knees, is counted within a maximum period of 30 s. Flexed arm hang (FAH) in s: The result represents the time taken for the strength activity, measured in tenths of a second. The stopwatch stops when the chin relaxes above the upper edge of the crossbar. Shuttle run (SR), i.e., natural running of a 10 × 5 m in s: the time taken to complete 5 full running cycles (10 m back and forth) is measured in tens of seconds.

### 2.6. Alternative Physical Education Program

The alternative program examined in this study is of a didactic–methodical nature and is directly related to the regular PE curriculum for third-grade elementary school students. It differs from the official standard program in the instructional units of athletics, gymnastics, and sports games, that cannot be implemented due to the inadequate material conditions in school gymnasiums, which have been replaced with instructional units of dance, folk dances, and gymnastics. A space of at least 40 square meters is designated, along with a minimum of two mats, bars, beams, ropes, a music device, and other small equipment.

Specifically, the difference in the implementation of the teaching content resulted from the alteration of 44 instructional units, or 43%, out of a total of 102 planned units annually in the regular PE curriculum (Table 1). The content replaced consisted of instructional units that were unfeasible to implement in the existing conditions, where the spatial and material–technical prerequisites were lacking. These included sports games and elements of gymnastics (vault, parallel bars, rings, and jumps). In their place, instructional units in dance, folk dances, and gymnastics were scheduled for implementation, for which the appropriate and optimal conditions existed. The remaining part of the alternative PE program, comprising 58 instructional units, or 57%, was identical to the rest of the regular PE program from which it was derived.

The proposed instructional units were tailored to the capacities of third-grade elementary school students. The teachers were trained to conduct these activities, and the distribution of lesson hours for instruction, practice, and assessment was appropriately arranged.

The most significant modifications in the alternative program compared to the standard PE curriculum pertained to the implementation of dance and folk dances. Out of the total of 44 modified instructional units during the six-month research period, 30 units were dedicated to dance and folk dances.

The use of dance and folk dances in the younger school age plays a crucial and meaningful role in shaping motor skills, as well as in the overall psychophysical development of children in that age group [21].

The core characteristic of dance activities is the integration of different forms of movement with musical accompaniment. Dance, folk dances, and social dances are performed with various steps, arm movements, and whole-body actions, along with musical accompaniment and a specific rhythm [22].

Dance activities contribute to the development of abilities related to recognizing, differentiating, and performing rhythmic structures, dynamics, tempo, spatial orientation, and control. Various forms of dance activities positively contribute to the development of coordination between the arms and legs, as well as the coordination of the entire body [23].

Notably, dance and folk dances significantly affect the acquisition of complex motor tasks, the reorganizing of movement stereotypes, rhythmic coordination, precision, and balance.

Many authors have underlined the significance and role of dance activities in fostering aesthetic education in school-age children [24,25].

The alternative PE program examined in this study encompasses the introduction of 14 instructional units in gymnastics, as well. These units, in terms of their content and characteristics, offer opportunities for implementation even under limited material and technical conditions within schools.

By respecting and acknowledging all the sensitive biological phases and laws related to human development during ontogenesis [26], we deduced that certain physical activities, based on their nature, character, and the complexity of their impact on individuals, exhibit varying degrees of adequacy within populations of diverse age groups, genders, health conditions, and similar factors. In this context, physical activities that facilitate the development of all motor and functional capacities are recommended for young school children.

Gymnastics stands out as an activity almost unparalleled in its diversity of movement. The exercises on apparatuses are of a polystructural, conventional, and acyclic nature [27].

While the repertoire of movements in this activity encompasses both acyclic and cyclic types, such as the approach when performing vaults or the approach for acrobatic diagonals on the floor exercise, the wealth of movements and exercises on the apparatuses empowers participants to build an extensive repertoire of motor knowledge. It fosters robust physical preparedness and the ability to respond effectively to everyday life situations [28]. For younger elementary school children covered by the alternative program, the extensive motor knowledge acquired is an excellent foundation for engagement in various sports activities [29].

### 2.7. Data Analysis

We used the R programming package (library pwr) in order to estimate the sample size. A multivariate analysis of covariance (MANCOVA) was employed to determine the effects resulting from the alternative PE program. A prerequisite for applying the multivariate analysis of covariance was to neutralize (equalize) the differences between the groups at the pre-intervention measurement. After achieving result neutralization, the real effects of the experimental program on the corresponding groups were identified. Parameters such as Wilk’s lambda, Rao’s R approximation, the degrees of freedom (df), and the significance level (*p*-level) were calculated. Inter-group differences at the univariate level with neutralization at the pre-intervention measurement were determined using a univariate analysis of covariance (ANCOVA), through adjusted means. The testing of differences was conducted through the F-test. The magnitude of the differences found was estimated by the effect size. The significance level was set at *p* ≤ 0.05. The data are reported as the minimum value, maximum value, range, and mean ± standard deviation, and were analyzed using the statistical package SPSS version 20.0 (SPSS Inc., Armonk, NY, USA) [30].

## 3. Results

The results of the basic descriptive statistics parameters for the motor abilities and the physical and health education of the participants in the experimental and control groups at the pre-intervention and post-intervention measurement are presented in Table 2 and Table 3.

Upon examining the results of the multivariate analysis of covariance (Table 4), which was applied to assess the anthropometric characteristics of body mass and height variables between the participants in the experimental and control groups at the post-intervention measurement, it can be concluded that there is no statistically significant difference either in body height or body mass (*p* = 0.143), nor in motor abilities or physical and health education (*p* = 0.368).

Figure 1 presents the change of mean values of observed variables for the experimental and the control group.

Table 5 presents the univariate differences for the applied variables between the participants in the experimental and control groups at the post-intervention measurement. A statistically significant difference was observed only for body height (*p* = 0.016). No statistically significant differences were found for the other applied measures.

Upon examining the results of the multivariate analysis of covariance (Table 6) which was applied to assess the motor abilities and sports–technical education among the boys and girls in the experimental and control groups at the post-intervention measurement, it can be concluded that no statistically significant difference was found at the multivariate level (*p* = 0.276 in boys and *p* = 0.405 in girls).

The results of the univariate analysis of covariance (ANCOVA) between the experimental and control groups of the boys and girls at the post-intervention measurement for motor skills and sports–technical education (Table 7) show that no statistically significant difference was observed for any variable. The differences between the two groups were found to be negligible (effect size, ES < 0.2).

## 4. Discussion

The results of the multivariate analysis of covariance applied to the assessed variables for the estimation of anthropometric characteristics, body mass, and height, indicate no statistically significant differences between the experimental group that underwent the alternative PE program and the control group at the post-intervention measurement. The implemented alternative program has led to certain, non-significant changes in the parameters for assessing growth and development, but a more detailed monitoring of the experimental program, in terms of intensity loads, is necessary to expect greater effects from the implemented PE classes [31]. Both the participants included in the experimental program and in the control group are in a phase of intensive growth and development so, besides the positive health outcomes that could be expected from the implementation of the specific PE program in the experimental group, changes in the anthropometric variables most likely occurred due to normal growth and development [32].

While sports are central to the PE curriculum, it is essential to recognize that sport and PE diverge in their core objectives, anticipated outcomes, teaching approaches, and overall purpose. Sport entails structured physical activities centered on formal competitions held on sports fields, often leading to narrowing opportunities, where success hinges solely on winning the game. Conversely, PE aims to equip younger school children with fundamental skills and essential knowledge about physical activities, prioritizing inclusivity and fostering healthy habits among a wide range of students as the measure of achievement in this subject. Success in PE is determined by the number of young students meeting these goals. However, this approach, favoring the majority or average, can impact both highly talented and struggling students. In sports, the primary objective, besides winning, lies in the honing and specializing of specific motor skills; whereas, in PE, these skills serve as a means to acquire practical motor knowledge [19].

The results of the multivariate analysis of covariance for assessing motor skills and physical–health education indicate an absence of statistically significant differences at the multivariate level (*p* = 0.368). The adjusted mean values of the univariate analysis of covariance indicate that the implemented alternative program led to improvements in the Flamingo balance test, the standing broad jump, and the sit-and-reach, but not at a statistically significant level, which could be attributed to the lack of proficiency in fundamental movement skills in children [33], the lack of appropriate material conditions [34], and insufficiently trained teaching staff [35,36].

The results of the univariate analysis of covariance for the variable of balance show that the experimental group that underwent the alternative PE program has better values for the Flamingo balance test compared to the control group (adjusted mean values), but without statistical significance (*p* = 0.317). Similar results were obtained by Milanović [37], who indicated that specially programmed PE classes positively impacted the motor skills of students in experimental groups. Stamatović and Šekeljić [38] aimed to determine the influence of PE on the motor status of participants, depending on the concept of implemented education for the following motor skills: explosive strength, repetitive strength, static force, sprint speed, segmental speed, flexibility, balance, coordination, and precision. The research represented a classic pedagogical experiment with parallel groups. The research results suggest that subject-oriented education influences improvement in most motor skills. Similar changes were identified in boys and girls from urban and rural schools in Italy [39]. In their study, Wright and associates [40] investigated the viability of a unique job-embedded professional development program spanning 10 weeks, which was aimed at enhancing teachers’ ability to provide enhanced PE lessons. However, it was observed that during its implementation by teachers, there was a decline in the motor skill proficiency of young school children, and the presence of potential confounding factors could not be discounted. The decrease in children’s movement skills might be attributed to the teachers’ emphasis on boosting participation and motivation, more deliberate lesson planning, and the use of seamless and engaging activities for transitioning children, instead of solely concentrating on skill refinement [40].

The results of the univariate analysis of covariance for segmental speed (hand tapping) show that there are no statistically significant differences (*p* = 0.834). The adjusted mean values indicate that greater changes were recorded for the experimental group, but they are not statistically significant. This suggests that the implemented alternative program, as well as regular PE classes, can lead to changes in segmental speed, which in our case were non-significant. Babin, Katić, and Vlahović [41] confirmed the impact of specially programmed PE on the frequency of movement or segmental speed, which is consistent with the present study. In the research by Stamatović and Šekeljić [38], the positive effects of the experimental program on the segmental speed of the experimental- group participants were also identified. The results of the study by Marković and Kopas-Vukašinović [42] are in line with the current research, with statistically significant differences for the values of the hand tapping test identified in favor of the students in the experimental groups.

Milanović [37] pointed out that programmed education has a positive impact on the motor skills of students in the experimental groups. Positive changes, compared to the results of the students in the control groups, were observed for the sprint-speed assessment tests. Through specifically programmed PE, a positive influence can be achieved on nearly all motor skills of school-age children, including movement frequency [41]. These results are comparable to those of Stojadinović, Zdravković, and Zdravković [43]. Their data show the positive impact of the experimental program and a statistically significant effect, which resulted in an increased number of correctly performed cycles of hand tapping (F = −23.386, *p* = 0.000).

Based on the aforementioned studies, it can be stated that the implemented alternative PE program resulted in positive changes in the segmental speed for the participants in the experimental group, with changes also observed in the participants of the control group, who were engaged in regular PE classes.

The results of the univariate analysis of covariance for flexibility (sit-and-reach test), demonstrated that no statistically significant differences were observed between the groups (*p* = 0.84). The adjusted mean values suggest that a greater change in flexibility was recorded among the participants in the control group. The positive effects of PE programs on changes in the flexibility of younger school-age students were established by several studies [37,38,41,44], which also emphasized the positive effects of executed exercise programs on the results of the sit-and-reach test in boys. The findings of the study conducted by Latino et al. (2021) [45] indicate that an online school-based exercise program, lasting approximately 60 min and occurring twice a week, with supervision by a PE teacher, could emerge as an effective strategy for enhancing flexibility (F1, 28 = 108.91, *p* < 0.001, η^2^p = 0.79, large effect size).

For the results obtained in this research, one of the causes may be the insufficient engagement of the participants in the experimental group during the post-intervention measurement. Furthermore, as flexibility has a negative correlation with growth and development [46], and decreases with age, one of the reasons for the results could be the developmental stage of the participants, as they are in a phase of intensive growth and development.

According to a recent review [47], investigations carried out in European regions have documented a consistent decline in the levels of flexibility in young school children. Conversely, in other global regions there have been reports of upward growth trends. For instance, in Canada, supplementary exercise programs have been implemented to address this concern.

Assessing the effects of the alternative program on explosive strength was conducted using the standing broad jump test. The results indicate that no statistically significant difference existed between the groups, but greater changes were noted in the participants in the control group. The obtained results are not consistent with the findings of research conducted by other authors.

Klinčarov, Nikovski, and Aceski [48] aimed to determine the impact of experimental treatments on the explosive leg strength of female students. The statistically significant impact of the program on the results of explosive leg strength in the female participants in the experimental group was established. Statistically significant differences regarding the increased values of the standing broad jump, in favor of the experimental group, were determined by Marković and Kopas-Vukašinović [42]. Similar results were obtained by other authors [32]. Rodić [49] concluded that systematic physical exercise in the experimental group had a significant positive influence on the development of students’ explosive strength, particularly in throwing and sprinting activities. Programmed PE had a positive impact on the throwing-type explosive strength in the participants of the experimental group compared to those in the control group [41].

The research conducted by Di Maglie and associates [50] indicates that a school-based intervention program spanning 6 months, which involve extracurricular physical activities with an added duration of 40 min per day for 5 to 6 days per week, has emerged as a successful approach for enhancing explosive strength (as demonstrated by the standing broad jump, with a significance level of *p* < 0.05).

Our results are most likely due to the insufficient intensity of load in the executed programs during PE classes. Researchers [51,52] have stated that the effects of exercise in children largely depend on the intensity during exercise, highlighting the need for greater attention to this aspect during PE classes.

Based on the results of the univariate analysis of covariance between the experimental and control groups at the post-intervention measurement, it can be observed that no statistically significant difference was established between the groups at the post-intervention measurement, either for the flexed arm hang test or for the handgrip strength test (*p* = 0.589 and *p* = 0.783, respectively). Although no statistically significant difference was found, the adjusted means were higher among the participants in the experimental group compared to the control group, indicating that the experimental program influenced changes in static strength for the participants in the experimental group.

Other researchers [38,42] have also shown the positive effects of PE programs on students’ static strength compared to students in the control groups. Babin, Katić, and Vlahović [41] identified the positive impact of the implemented program on the static arm- and shoulder-strength of students in the specifically programmed PE. The positive trends in the development of all motor skills in both genders after the execution of specific PE programs were studied by Jurak, Kovač, and Strel [44]. The authors determined that the effects of the program itself indicated better results for most of the tests of the experimental group compared to the control group for boys, including a change in the flexed arm hang test.

The performance of pull-up endurance increased significantly (flexed arm hang, F = 28.82, *p* < 0.01, η^2^ = 0.33) in eight-year-old girls who participated in an eight-week integrated neuromuscular training two times per week, within the first ~15 min of PE class [53].

The obtained research results of this study, in addition to those of priorly conducted research, suggest that modifying the existing curricula to a lesser extent could influence the improvement of static strength and pull-up endurance for the participants engaged in these programs. However, grip strength improves as age increases, regardless of gender [54].

The obtained values of the univariate analysis of covariance for sprint speed show that no statistically significant difference existed between the experimental and control groups (*p* = 0.623). Better values were recorded for the control group, although the adjusted mean values differed very slightly numerically. In this case, lower values represent better results. In previous studies [55,56], no improvement in speed was identified after the experimental treatment. The authors state that such results were expected, given that speed, as a motor skill, is significantly genetically determined. Additionally, the experimental period was relatively short to anticipate significant transformations in speed among the students in both groups. Nevertheless, certain studies have demonstrated that appropriately programmed PE classes can have positive effects on speed. Research findings [37] have indicated that programmed PE can influence students’ motor skills and bring about positive changes in sprint-speed assessment tests. Researchers [49] have concluded that physical exercise contributes to the development of explosive strength, especially in throwing and sprinting activities. Other researchers have also confirmed the positive effects of PE classes on sprint speed [32,35]. Researchers [11] showed improvements in a 20 m shuttle run in the experimental group of school children aged 10.5 ± 0.5 years, but the authors themselves were not sure if the improvements were simply a consequence of the higher volume of endurance activities the experimental group underwent as compared to the control group (the weekly physical activity volume increased by about +21% during the 12 weeks of the program, whereas no change in physical activity volume was noted for the control group). As most of the mentioned studies were for experimental programs lasting for an extended period of time, the obtained results are most likely due to this duration, or due to typical biological growth and development [46].

Regarding the differences in the control and experimental groups divided by gender, no significant difference was found. A difference was found only in body height, which is a body-composition parameter, and was not influenced by the program. This confirms our results in more detail and shows that both genders progressed equally. The strength of these results can be found in the age of the participants, since the maturation process had not started for either genders [57].

The obtained values of the univariate analysis of covariance for physical and health education showed that no statistically significant difference existed between the experimental and control groups (*p* = 0.220). Based on the obtained data, it can be stated that the experimental group achieved greater results, but in comparison to the control group, this difference is almost negligible numerically.

Šekeljić [58], in light of the results of his research, pointed out that the use of educational content incorporating elements of sports games, specifically basketball, can effectively influence the achievement of PE objectives, such as children’s health, cognitive functions, and overall physical fitness, in the context of acquiring motor skills and physical–health education.

The significance of our study lies in its exploration of whether an alternative PE program could effectively replace the conventional PE curriculum for younger elementary school students, while still aligning with the goals and objectives of PE. Since both programs in our study contributed to the physical development of the children equally, future studies should focus on the duration and intensity of PE classes, and how they affect potential improvements in the overall health of children. This implies that more time, and with a greater intensity, dedicated to PE may lead to greater improvements in motor abilities and health education.

## 5. Study Limitations

The limitations of this study include the bias of participants selection. This is a study conducted in an elementary school with modified PE classes, and conditions did not allow any type of randomization of the participants that could increase the strength of the methodology. Although the duration of the PE classes is mentioned, no information about the intensity of the exercises and activities is given in this research. Additional information on intensity would be beneficial for a more comprehensive discussion. However, this study included results for both sexes, which provides gender-related generalizability.

## 6. Conclusions

The results obtained from the conducted research indicate the following practical conclusions: Both the alternative PE program and the regular PE program had positive effects on motor and physical and health education to a limited extent. Namely, these positive effects were not statistically significant at either the multivariate (considering multiple variables together) or univariate (considering individual variables) levels; the alternative PE program proposed in this study, with modifications to the structure of the existing, regular PE program, can fully replace regular PE classes in schools that do not meet the required spatial and material standards. This suggests that the alternative PE program could be a viable option for schools facing limitations to resources.

## Figures and Tables

**Figure 1 children-10-01657-f001:**
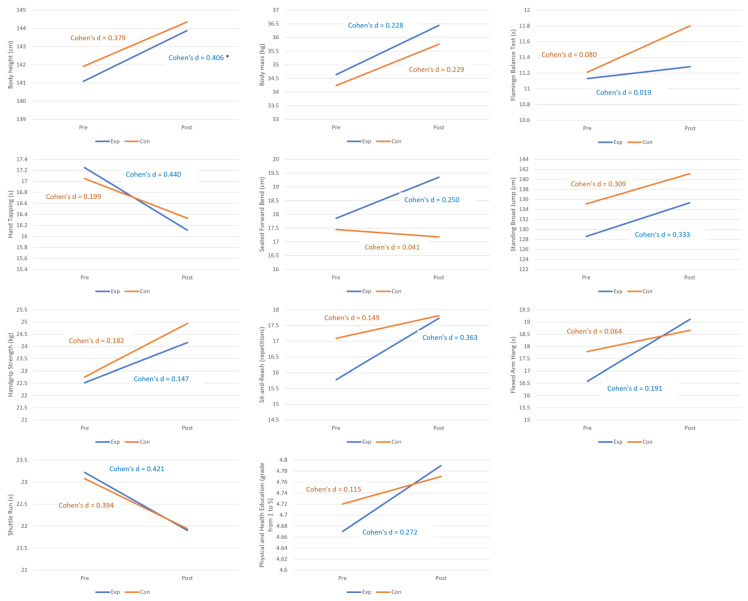
Change of mean values of observed variables for the experimental and the control group (* significant difference between experimental and control group). Legend: Pre—pre-intervention; Post—post-intervention; Exp—experimental group; Con—control group.

**Table 1 children-10-01657-t001:** Comparative overview of the educational units (1–102) by month (SEPTEMBER–MAY).

Regular PE Program	Alternative PE Program
SEPTEMBER: 1. Elementary games of choice by students; 2. throwing and catching the ball in various ways; 3. walking and running at different paces; 4. fast running with a change of position from various starting positions; 5. fast running; 6. fast running for 40 m from various starting positions; 7. endurance running in nature; 8. long jump from a marked take-off point; 9. long jump; 10. endurance running in nature; 11. long jump from a marked take-off point; 12. skipping rope in pairs; 13. the elementary games including fast running. OCTOBER: 14. Throwing a ball for distance; 15. throwing a ball at a moving target; 16. walking and running at different paces; 17. elementary games with shooting at moving targets; 18. the elementary game between two fires; 19. high jump of 1 m; 20. lifting and carrying a third person on a pole; 21. high jumps with a soft landing at a height of 1 m; 22. the elementary game between two fires; 23. dribbling a ball with one hand while running straight; 24. dribbling a ball while running with one hand; 25. rolling a hoop in a circle and zigzagging; 26. dribbling a ball with one hand while running; 27. rolling a hoop in a circle and zigzagging. NOVEMBER: 28. Crawling on the belly, on the side, and on the back while bypassing obstacles; 29. elementary games for developing abilities; 30. crawling in different ways while bypassing obstacles and carrying light objects; 31. high jump with a straight approach; 32. high jump with a straight approach; 33. pulling and pushing in pairs across a line; 34. high jump with a straight approach up to 50 m high; 35. pulling and pushing in pairs; 36. passing a ball in pairs while running; 37. passing a ball in pairs while running; 38. passing a ball while running in columns, rows, and circles. DECEMBER: 39. Pulling in a circle while holding hands (skipping rope); 40. crawling and climbing on ladders and through windows; 41. pulling and pushing a rope in a square shape; 42. forward roll from a squat to a squat; 43. forward roll from a squat to a squat; 44. aesthetic body shaping—exercise for the shoulder girdle and arms; 45. forward roll from a squat to a squat; 46. free composition on the floor; 47. aesthetic body shaping; 48. “The White Vine Has Bent”—a children’s dance; 49. “The White Vine Has Bent”—a children’s dance; 50. aesthetic body shaping and the continued practice of folk dance; 51. aesthetic body shaping and the continued practice of folk dance. JANUARY: 52. Elementary games fostering collaborative relationships; 53. rhythmic movement emphasizing individual parts of the 2/4 and 4/4 beats; 54. walking in a group forward and backward with a cone on a low beam; 55. walking in a group on a low beam with half and full turns; 56. relay games with elements of agility and balance. FEBRUARY: 57. Walking in a group on a low beam; 58. pulling and pushing a third person on a sled; 59. rhythmic movement in circles, zigzags, and their combination; 60. pulling and pushing a third person on a sled; 61. dance steps, the gallop, straddle, and forward–backward; 62. free composition on a low beam; 63. free improvisation with a galloping step; 64. free composition on a low beam; 65. the elementary game “Remember the Objects”; 66. lifting and carrying a third person on a pole; 67. ball- bouncing “volleyball” in a circle. MARCH: 68. Lifting and carrying a third person on poles; 69. ball-bouncing “volleyball” with a partner; 70. crossing from apparatus to apparatus; 71. exercises with pole manipulation while in pairs; 72. crossing from an apparatus to an object; 73. folk dance from Serbia; 74. the elementary game “volleyball” by bouncing; 75. folk dance; 76. folk dance; 77. running in nature at a varying pace; 78. jumping over a short rope while running; 79. jumping over a short rope while running; 80. “rim” running. APRIL: 81. Climbing ropes using ropes and ladders; 82. climbing ropes using ropes and ladders; 83. front vault, over knee height; 84. walking on stilts forward–backward and sideways; 85. front vault, over knee height; 86. climbing ropes, using ropes and sailor ladders; 87. “Between Four Fires,” an elementary game; 88. climbing ropes using ropes and ladders; 89. fast running for 40 m; 90. obstacle course with tasks—jumping, climbing, and crawling. MAY: 91. Fast running for 40 m; 92. elementary games of choice by the students; 93. hurdle race; 94. kicking a ball with the foot; 95. standing long jump; 96. passing under a long rope in groups; 97. the elementary game between two and four fires; 98. jumping over a long rope in groups; 99. fast running for 40 m; 100. lifting and carrying a third person on logs; 101. jumping over a long rope in groups; 102. elementary games.	SEPTEMBER AND OCTOBER: 1–27. The alternative PE program is identical to the regular PE program. NOVEMBER: 28. Forward roll; 29. forward roll; 30. backward roll; 31. backward roll; 32. forward roll; 33. forward roll; 34. high jump with a straight approach; 35. high jump with a straight approach; 36. high jump with a straight approach; 37. basic elements of dance techniques and their rhythmic structures; 38. basic elements of dance techniques and their rhythmic structures. DECEMBER: 39. Basic elements of dance techniques and their rhythmic structures; 40. Branko’s circle dance; 41. Branko’s circle dance; 42. King’s circle dance; 43. King’s circle dance; 44. pulling in a circle holding hands; 45. pulling and pushing in a square shape; 46. aesthetic body shaping—exercise for the shoulder girdle; 47. aesthetic body shaping; 48. “The White Vine Has Bent”—a children’s dance; 49. “The White Vine Has Bent”—a children’s dance; 50. aesthetic body shaping and the continued practice of folk dance; 51. rhythmic movement emphasizing the individual parts of the 2/4 and 4/4 beats. JANUARY: 52. Frog jump; 53. frog jump; 54. handstand; 55. handstand; 56. walking on a low beam. FEBRUARY: 57. Walking on a low beam; 58. frog jump, handstand, and walking on a low beam; 59. “Moravac” dance 1, 2; 60. “Moravac” dance 1, 2; 61. “Kačerac” dance basics; 62. “Kačerac” dance basics; 63. free composition on a low beam; 64. free composition on a low beam; 65. the elementary game “Remember the Objects”; 66. lifting and carrying a third person on a pole; 67. English waltz basics. MARCH: 68. English waltz basics; 69. English waltz right turn; 70. English waltz right turn; 71. exercises with pole manipulation while in pairs; 72. jumping over a short rope while running; 73. jumping over a short rope while running; 74. “I Planted Flax” (folk dance); 75. “I Planted Flax” (folk dance); 76. “I Planted Watermelon” (folk dance); 77. “I Planted Watermelon” (folk dance); 78. forward roll; 79. backward roll; 80. frog jump. APRIL: 81. Handstand; 82. “Early Quinces” (folk dance); 83. “Early Quinces” (folk dance); 84. “Kolo Vodi Vasa” (folk dance); 85. “Kolo Vodi Vasa” (folk dance); 86. folk dances; 87. Spinko waltz 2; 88. Spinko waltz 2; 89. social games; 90. exercises in pairs. MAY: 91. Fast running for 40 m; 92. elementary games of choice by the students; 93. hurdle race; 94. kicking a ball with the foot; 95. standing long jump; 96. passing under a long rope in groups; 97. the elementary games between two and four fires; 98. jumping over a long rope in groups; 99. fast running for 40 m; 100. lifting and carrying a third person on logs; 101. jumping over a long rope in groups; 102. elementary games.

**Table 2 children-10-01657-t002:** Basic descriptive parameters of motor abilities and sports–technical education of participants in the experimental group at pre-intervention and post-intervention measurement (n = 105).

	Measure	Mean	Min	Max	Range	SD
Body Height (cm)	Pre	141.09	121.00	156.00	35.00	6.40
	Post	143.87	126.00	167.00	41.00	7.28
Body Mass (kg)	Pre	34.64	21.10	66.00	44.90	7.60
	Post	36.44	23.00	68.00	45.00	8.16
Flamingo Balance Test (s)	Pre	11.13	1.50	37.40	35.90	7.88
	Post	11.28	1.00	38.20	37.20	7.97
Hand Tapping (s)	Pre	17.25	12.08	29.00	16.92	2.79
	Post	16.11	11.70	21.20	9.50	2.37
Seated Forward Bend (cm)	Pre	17.86	5.00	38.00	33.00	5.43
	Post	19.35	0.50	38.00	37.50	6.44
Standing Broad Jump (cm)	Pre	128.57	85.00	172.00	87.00	20.29
	Post	135.31	95.00	180.00	85.00	20.16
Handgrip Strength (kg)	Pre	22.52	5.00	62.00	57.00	10.84
	Post	24.16	0.00	55.00	55.00	11.46
Sit-and-Reach (repetitions)	Pre	15.78	4.00	29.00	25.00	5.47
	Post	17.73	5.00	29.00	24.00	5.27
Flexed Arm Hang (s)	Pre	16.58	0.00	61.50	61.50	13.05
	Post	19.11	2.00	65.00	63.00	13.38
Shuttle Run (s)	Pre	23.22	17.03	31.22	14.19	3.24
	Post	21.90	16.19	31.12	14.93	3.02
Physical and Health Education (grade from 1 to 5)	Pre	4.67	4.00	5.00	1.00	0.47
	Post	4.79	4.00	5.00	1.00	0.41

Legend: Measure—measurement; Pre—pre-intervention; Post—post-intervention; Mean—mean value; Min—minimum value; Max—maximum value; SD—standard deviation.

**Table 3 children-10-01657-t003:** Basic descriptive parameters of motor abilities and sports–technical education of participants in the control group at pre-intervention and post-intervention measurement (n = 109).

	Measure	Mean	Min	Max	Range	SD
Body Height (cm)	Pre	141.91	127.00	160.00	33.00	6.35
	Post	144.35	130.00	166.00	36.00	6.53
Body Mass (kg)	Pre	34.24	22.30	59.00	36.70	6.43
	Post	35.75	24.10	59.00	34.90	6.77
Flamingo Balance Test (s)	Pre	11.21	1.00	38.10	37.10	7.35
	Post	11.80	1.00	40.30	39.30	7.46
Hand Tapping (s)	Pre	17.05	10.36	37.44	27.08	3.38
	Post	16.33	10.98	44.47	33.49	3.85
Seated Forward Bend (cm)	Pre	17.45	0.00	33.00	33.00	6.38
	Post	17.18	0.40	34.00	33.60	6.77
Standing Broad Jump (cm)	Pre	135.08	80.00	178.00	98.00	18.75
	Post	141.14	90.00	194.00	104.00	20.46
Handgrip Strength (kg)	Pre	22.75	4.00	55.00	51.00	11.61
	Post	24.94	0.50	50.00	49.50	12.44
Sit-and-Reach (repetitions)	Pre	17.09	6.00	26.00	20.00	4.69
	Post	17.81	3.00	28.00	25.00	4.98
Flexed Arm Hang (s)	Pre	17.79	0.00	61.00	61.00	13.33
	Post	18.66	1.00	62.50	61.50	13.81
Shuttle Run (s)	Pre	23.08	17.30	33.55	16.25	3.03
	Post	21.94	17.20	30.38	13.18	2.75
Physical and Health Education (grade from 1 to 5)	Pre	4.72	4.00	5.00	1.00	0.45
	Post	4.77	4.00	5.00	1.00	0.42

Legend: Measure—measurement; Pre—pre-intervention; Post—post-intervention; Mean—mean value; Min—minimum value; Max—maximum value; SD—standard deviation.

**Table 4 children-10-01657-t004:** Multivariate analysis of covariance (MANCOVA) between experimental and control groups at post-intervention measurement.

	Wilk’s	F	df1	df2	*p*
Body Height and Body Mass	0.982	1.96	2	209	0.143
Motoric and Physical and Health Education	0.921	1.09	11	180	0.368

Legend: Wilk’s—Wilk’s lambda test; F—Rao’s F approximation; df—degrees of freedom; *p*—significance level.

**Table 5 children-10-01657-t005:** Univariate analysis of covariance (ANCOVA) between experimental and control groups at post-intervention measurement.

	Adj Means		F	*p*-Level
	Con.	Exp.		
Body Height (cm)	144.59	145.28	5.91	0.016 *
Body Mass (kg)	35.95	36.37	2.30	0.131
Flamingo Balance Test (s)	12.43	11.75	1.00	0.317
Hand Tapping (s)	16.13	16.07	0.04	0.834
Seated Forward Bend (cm)	18.35	18.63	0.18	0.668
Standing Broad Jump (cm)	141.70	139.39	1.62	0.204
Handgrip Strength (kg)	24.93	25.56	0.29	0.589
Sit-and-Reach (repetitions)	17.81	17.71	0.04	0.840
Flexed Arm Hang (s)	20.18	20.43	0.07	0.783
Shuttle Run (s)	21.67	21.54	0.24	0.623
Physical and Health Education (grade from 1 to 5)	4.75	4.81	1.51	0.220

Legend: Adj Means—Adjusted means; Con.—control group; Exp.—Experimental group; F—Rao’s F approximation; *p*-level—significance level; statistical significance * *p* < 0.05.

**Table 6 children-10-01657-t006:** Multivariate analysis of covariance (MANCOVA) between experimental and control groups of boys and girls.

Gender	Wilk’s	F	df1	df2	*p*
Males	0.830	1.25	11	67	0.276
Females	0.813	1.07	11	51	0.405

Legend: Wilk’s—Wilk’s lambda test; F—Rao’s F approximation; df—degrees of freedom; *p*—significance level.

**Table 7 children-10-01657-t007:** Univariate analysis of covariance (ANCOVA) between the experimental and control groups of boys and girls at post-intervention measurement.

	Gender	Adj Means		F	*p*-Level	ES
		Con.	Exp.			
Body Height (cm)	Males	145.36	145.50	1.39	0.242	0.053
	Females	144.62	143.55	5.40	0.024 *	
Body Mass (kg)	Males	36.24	36.51	0.43	0.513	0.041
	Females	36.08	35.49	2.48	0.120	
Flamingo Balance Test (s)	Males	11.85	10.67	2.63	0.109	−0.058
	Females	13.03	13.07	0.01	0.974	
Hand Tapping (s)	Males	16.49	16.35	0.15	0.698	−0.135
	Females	15.64	15.80	0.12	0.726	
Seated Forward Bend (cm)	Males	17.55	16.97	0.39	0.532	0.296
	Females	20.74	19.30	2.85	0.097	
Standing Broad Jump (cm)	Males	143.12	142.63	0.05	0.828	0.035
	Females	135.83	139.77	1.83	0.181	
Handgrip Strength (kg)	Males	26.24	28.12	2.29	0.135	−0.049
	Females	22.61	23.09	0.06	0.807	
Sit-and-Reach (repetitions)	Males	17.70	17.58	0.03	0.872	0.241
	Females	18.03	17.84	0.08	0.782	
Flexed Arm Hang (s)	Males	21.61	20.68	0.54	0.464	0.125
	Females	20.06	18.36	1.48	0.228	
Shuttle Run (s)	Males	21.74	21.71	0.01	0.913	−0.057
	Females	21.31	21.62	0.41	0.525	
Physical and Health Education (grade from 1 to 5)	Males	4.70	4.81	2.99	0.088	0.152
	Females	4.82	4.83	0.09	0.766	

Legend: Adj Means—adjusted arithmetic means; Con.—control group; Exp.—experimental group; F—Rao’s F approximation; *p*-level—significance level; statistical significance * *p* < 0.05; ES—effect size.

## Data Availability

Not applicable.

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
