# Peer review of "Efficiency of an Alternative Physical Education Program for the Lower Grades of Elementary School Children"

_children, 2023, doi:10.3390/children10101657_

Round 1
Reviewer 1 Report
The article contains an interesting introduction supported by a broad review of the literature. It is commendable that the most recent position was used. Main comments:
- Add how the sample size was calculated
- Consider the use of a flow chart
-Shorten the methods section
- add 0 to the p-value for overall clarity
- specify limitations in a separate section
- Add practical conclusions
Author Response
Dear,
we would like to thank the reviewer for contributing to the substantial improvement of our manuscript.
Please find our responses attached.
Kind regards,
the authors

Reviewer 2 Report
Dear authors,
Thank you very much for your work. However, I do have several questions. Please find my specific comments below:
Abstract:
1. Effect size interpretations are needed in the abstract
Introduction:
2. Lines 51 to 65 suggest merging into one paragraph and condensing the text.
3. There are several paragraphs in the study that have only one sentence, making the expression of the theme unclear. Also, similar descriptions appear below, e.g., lines 192-193, 197-198, 199-200, 230-232, 243-244, etc. It is recommended that multiple sub-paragraphs on a theme be combined.
4. It is recommended that the status of existing relevant research be supplemented. For example, is there any existing research that has been written on content reforms for status quo issues such as poor infrastructure conditions. If so, it is recommended to add descriptions (What was the content of the reform? What is the effect of the reform? etc.).
Methods:
5. Please note that MDPI Instructions for authors the editorial adheres to the CONSORT statement in clinical trials. MDPI requires a completed CONSORT 2010 checklist and flow diagram as a condition of submission when reporting the results of a randomized trial. Templates for these can be found here or on the CONSORT website (http://www.consort-statement.org) which also describes several CONSORT checklist extensions for different designs and types of data beyond two group parallel trials. At minimum, your article should report the content addressed by each item of the checklist. It is imperative to describe the randomization process.
Results:
6. Tables I and II are proposed to be combined to report for the within-group difference test for the experimental and intervention groups, increasing the reporting of difference values (Cohen’s d).
7. Body Mass in Table 6 shows a statistically significant difference (p=0.024), yet it is not labeled "*".
8. It is recommended that differences within and between groups be graphed. Authors can refer to the following figures.
Discussion:
9. It is recommended to add a description of the significance of the study and the limitations of the study.
Author Response

(The authors gave the same response as above.)

Reviewer 3 Report
Congratulations to the authors for the article.
In it, a 6-month intervention with an alternative physical education programme has been carried out in 214 third graders.
This study is interesting in its approach but the article needs some improvement.
It is difficult for the reader to get an idea of what the differences are between the two approaches to physical education.
The introduction needs reorganisation. For example, it starts describing the problems of physical education in Serbia, but it does not really clarify the evidence for choosing the alternative programme design approaches.
It would be desirable to improve the objectives of the study and to clarify these aspects in the summary and discussion.
The age of the subjects should be clearly indicated so that the international reader can identify the age of the subjects.
The creation or design of a table or image to compare the characteristics of the two programmes would be helpful.
In methodology, the alternative programme (line 206) should be described and the evidence justifying it should be part of the introduction.
The discussion uses a large number of quotations that have not been considered in the introduction, which is inappropriate as the discussion should be based on the points raised in the introduction.
The conclusions should be more in line with the results and the discussion.
Author Response

(The authors gave the same response as above.)

Round 2
Reviewer 2 Report
The revised version is better. But there are still some details of the article that need further improvement, for example, the figure 1 needs to be expressed more scientifically and standardized.
Reviewer 3 Report
Congratulations to the authors for the substantial improvements they have made to the article. I consider that the doubts and problems raised have been clarified.